# Feel Good, Eat Better: The Role of Self-Compassion and Body Esteem in Mothers’ Healthy Eating Behaviours

**DOI:** 10.3390/nu13113907

**Published:** 2021-10-30

**Authors:** Noémie Carbonneau, Anne Holding, Geneviève Lavigne, Julie Robitaille

**Affiliations:** 1Department of Psychology, Université du Québec à Trois-Rivières, Trois-Rivières, QC G8Z 4M3, Canada; genvievelavigne@hotmail.com; 2Department of Psychology, New York University, New York, NY 10003, USA; anne.holding@mail.mcgill.ca; 3Nutrition, Health and Society (NUTRISS) Research Center, School of Nutrition, Université Laval, Quebec City, QC G1V 0A6, Canada; julie.robitaille@fsaa.ulaval.ca

**Keywords:** self-compassion, body esteem, intuitive eating, emotional eating, diet quality

## Abstract

Mothers’ eating behaviours are important to ensure the health and well-being of themselves and their families. Recent research has pointed to self-compassion, defined as extending kindness to oneself in times of perceived inadequacy or general suffering, as a trait associated with healthy forms of eating, such as intuitive eating, and reduced maladaptive forms of eating, such as emotional eating. However, little is known about the psychological mechanism through which self-compassion relates to healthy eating behaviours. This study examined 100 mothers’ levels of self-compassion, body esteem and eating behaviours. Structural equation modelling revealed that self-compassion was positively associated with diet quality and intuitive eating, while being negatively associated with emotional eating. Moreover, these links occurred, in part, due to higher body esteem. This points to a mechanism through which self-compassion may positively contribute to mothers’ healthy eating behaviours. The implications for eating outcomes and women’s health are discussed.

## 1. Introduction

There is increasing interest in the potential benefits of self-compassion—a kind approach toward oneself during personally challenging times [1,2]—as a promising approach to promoting psychological and physical well-being. For example, self-compassion has been shown to relate to greater life satisfaction as well as less anxiety and depression [1,3,4,5]. There is also evidence that a compassionate attitude toward oneself could promote health-protective practices, such as healthier eating habits, exercise, sleep behaviours, and stress management (e.g., [6,7]). In addition, increasing research suggests that self-compassion protects against a negative body image and eating disorders in adults (e.g., [8,9]). Yet, positive body image is more than simply not having a maladaptive body image [10], and a positive approach to food is more than simply the absence of eating pathology [11]. Therefore, more research is needed to examine how a self-compassionate attitude may promote positive, healthful relationships with food and body image. It appears especially valuable to examine such questions among mothers, who often feel the responsibility to model and transmit positive eating and body image attitudes to their children [12,13] while simultaneously grappling with these issues personally (e.g., [14,15]). Thus, expanding the existing research, the present study aims to examine the associations of self-compassion with intuitive eating, emotional eating, and diet quality in a sample of mothers. In addition, we propose a mechanism through which self-compassion is associated with diet quality, intuitive eating, and emotional eating, namely, body esteem. The implications for eating outcomes and women’s health are discussed. 

### 1.1. Self-Compassion

Self-compassion is a general positive attitude characterized by the ability to relate to one’s feelings of suffering with warmth and understanding. Self-compassion implies offering nonjudgmental understanding of one’s shortcomings, and recognizing that painful emotions and experiences are part of the human nature [2]. Neff has defined self-compassion as being composed of three main interrelated components: self-kindness, mindfulness, and common humanity [1,2]. Self-kindness refers to being supportive and gentle, rather than harsh and critical toward ourselves when we fail or suffer. Mindfulness refers to acknowledging (rather than ignoring) one’s suffering, and being conscious of negative thoughts or feelings, without becoming “overidentified” with them [2]. Finally, common humanity is the capacity to recognize that personal failures and feelings of inadequacies are shared by all, and that we are therefore not alone in our struggles.

Self-compassion has been identified as a protective factor against negative body image and eating pathology (e.g., [8,16]). For example, longitudinal research over a four-month period has shown that higher self-compassion could protect girls against declines in body satisfaction and increases in eating pathology [17]. Self-compassion has also been positively linked with more health-promoting behaviours in a sample of mothers [18]. In addition, research suggests that self-compassion could help preserve women’s body appreciation during body-related threats [19,20]. Past research has also shown that self-compassion is positively related to body esteem (e.g., [12,21]), an association that the present research aims to replicate. 

Self-compassion has been shown to promote a relationship with food based on honoring the body’s needs, as evidenced by its positive association with intuitive eating [22] and its negative association with emotional eating [12]. However, the association between self-compassion and diet quality is less clear. Higher self-compassionate individuals have been found to display less motivation to eat palatable but unhealthy foods [23]. There is also evidence that daily fluctuations in self-compassion could play a role in the quality of the food eaten [24]. Yet, in a study conducted by Mantzios, Egan, Hussain, Keyte, and Bahia [25], self-compassion was not found to be significantly related to fat and sugar consumption. More research on the association between self-compassion and diet quality is therefore needed. 

Research by Schoenefeld and Webb [22] suggests that body image acceptance and action (i.e., cognitive flexibility and acceptance relative to body image) mediates the association between self-compassion and intuitive eating. Moreover, Carbonneau and colleagues [12] reported evidence of the mediating role of body esteem in the relationship between self-compassion and emotional eating. This recent body of evidence suggests that holding a positive body image may be the pathway through which those with high self-compassion experience more positive eating behaviours and relationships with food. 

### 1.2. Body Esteem

Individuals who have a positive body image have been shown to demonstrate higher care and respect towards their body, which should be associated with healthier eating behaviours as well as the selection of high-quality foods [26,27,28,29]. In the present research, we examine a specific facet of positive body image, namely, body esteem, that represents positive self-evaluations about one’s appearance and weight (e.g., [30,31]). We hypothesize that body esteem should be positively related to intuitive eating and diet quality, and negatively related to emotional eating. These hypotheses are based on previous research showing that positive body image predicts health-protective behaviours and eating patterns. For example, a number of studies have reported that positive body image relates to intuitive eating (e.g., [26,32,33,34,35]). In the same vein, research also shows that higher body esteem relates to lower levels of emotional eating [12]. While research is scarce on the association between body esteem and diet quality, there is evidence that body image dissatisfaction relates to unhealthy eating habits. More specifically, in a study conducted in Portugal, women dissatisfied with their body image were found to consume more ultra-processed foods as well as less unprocessed (or minimally processed) foods than their body image satisfied counterparts [36]. A five-year longitudinal study found that low positive body image was associated with more unhealthy diets and eating behaviours, as well as less exercise, even after considering the influence of body mass index (BMI [37]). 

In this paper, we posit that self-compassion will be directly and/or indirectly (via body esteem) related to eating behaviours and diet quality. Indeed, research suggests that daily fluctuations in self-compassion play a role in the quality of the food that is eaten [24] and in the levels of intuitive eating [9]. Other researchers have reported that self-compassion is strongly associated with fewer feelings of shame toward one’s body [38] and helps mitigate the negative effects of social media on body image [39]. In addition, individuals with eating disorders have been shown to display lower levels of self-compassion and mindfulness [40]. Self-compassion also appears to protect individuals from the negative effects of appearance contingent on self-esteem and social comparisons [19,41,42].

### 1.3. Eating Behaviours

Emotional eating is a common unhealthy eating behaviour that has been shown to be associated with elevated BMI [43,44,45] and has been suggested as a predictor of eating disorders such as binge eating [46,47]. Emotional eating occurs when individuals eat in response to a negative emotional state such as anxiety or sadness instead of in response to dietary needs [48,49]. Emotional eating has been associated with an inability to regulate one’s emotions [50,51] and is believed to be an avoidant coping strategy aimed at reducing negative emotions [52,53]. 

Conversely, intuitive eating, defined as the use of physiological cues to determine when, what, and how much to eat while maintaining a positive relationship with food [54], has gained a lot of research attention in recent years. Intuitive eating involves eating behaviours based on hunger and satiety signals, assuming that the body naturally knows what it needs to thrive [55]. Intuitive eating has been associated with lower BMI, positive eating behaviours, and well-being (e.g., [55,56,57,58,59,60]). Individuals who display low levels of intuitive eating appear to also display a number of unhealthy behaviours such as food addiction, dietary restraint, emotional eating, and compensatory weight control behaviours (e.g., [56]). 

### 1.4. Studying Eating Behaviours and Body Image in Women and Mothers

This research was conducted in women, since dietary restraint and emotional eating are prevalent female health issues [61]. More specifically, participants in our study were all mothers. Motherhood can be a challenging time for body image. Lasting changes from pregnancy and childbirth (e.g., weight retention, stretch marks, or scars) can be perceived negatively and lead to decreased body satisfaction [62,63], especially in societies that pressure mothers to rapidly regain their pre-pregnancy bodies. These pressures to lose the “baby weight” have been associated with maladaptive eating patterns in mothers (e.g., [64]). Yet, motherhood can also have positive effects on body image and eating behaviours. For example, many women report that pregnancy has made them more aware of the importance of healthy eating [65]. In addition, it is not uncommon for new mothers to report greater respect for and amazement at their body’s capabilities [66,67]. Motherhood can also motivate women to be kinder with their body and talk about it in gentler terms as they want to role model a positive body image [68]. Mothers indeed play a critical role in their children’s current and future body image and eating behaviours and have been shown to transfer their own attitudes and behaviours related to food and body image to their children [69,70,71,72]. Extensive evidence exists for the transmission of parents’ weight and shape concerns to their children (e.g., [73,74,75,76]). Recently, Hart, Tan, and Chow [77] showed that mothers with anti-fat attitudes have daughters who display more dietary restraint. An experimental manipulation where mothers were instructed to make negative comments about their own appearance while with their 8- to 12-year-old daughters was shown to reduce young girls’ body esteem and body appreciation as well as to lead to a greater consumption of sugary food post-experimentation [71]. Research also points to the transmission of positive body image and healthy eating behaviours. For instance, the children of parents who eat healthier foods, such as fruits and vegetables, are likely to eat more of these foods [78]. Together, these results suggest that the attitudes and behaviours of mothers with regard to body image can have significant consequences for their own health, as well as a determining influence on their children’s body esteem, body satisfaction, attitudes related to body image, diet, and actual eating behaviours. Thus, it is important to understand the antecedents of healthy (and unhealthy) eating patterns in mothers. 

### 1.5. The Present Research

In the present study, we posit that women’s general attitude of acceptance of themselves should be associated with positive eating behaviours, such as listening to one’s hunger and satiety cues and eating high-quality foods that support health and well-being. In addition, we posit that body esteem will act as a mediator of these relationships. Thus, based on previous literature, it is hypothesized that self-compassion will be positively related to body esteem, which, in turn, will be positively related to diet quality and intuitive eating as well as negatively related to emotional eating (see Figure 1).

## 2. Methods

### 2.1. Participants and Procedure

Participants were recruited through a Facebook ad targeting women living in the province of Quebec (Canada) and who had at least one child aged between 2 and 8 years old. No additional inclusion criteria were specified. Recruitment took place between September and December 2018. All questionnaires were completed online through an online survey platform. All participants read and signed the consent form. The study was conducted according to the guidelines of the Declaration of Helsinki and approved by the Ethics Committee of Université du Québec à Trois-Rivières (protocol code CER-18-244-07.10; obtained April 5th, 2018). The present study was part of a larger project investigating mothers’ attitudes and behaviours related to eating and body image. The final sample consisted of participants who completed all the measures relevant to the present study.

The final sample included 100 French-Canadian mothers with a mean age of 33.70 (SD = 4.48, 73% between 30 and 38 years old). They had between 1 and 5 children (M = 2.07; SD = 0.86) who were between 1 month and 16 years old. The majority of the participants had a university-level education (32% Bachelors’ degree and 25% Masters’ or Doctorate degree) and a household income of more than $75,000 CAD (63.5%). All participants identified as White. A total of 47% of participants had a full-time employment, 21% were on maternity leave and 10% were homemakers. 

### 2.2. Measures

Demographic Variables: A number of demographic questions were asked such as age, education, employment, and ethnicity.

Body Mass Index: Participants reported their current weight and height. These were used to compute their BMI. More specifically, BMI was calculated using participants’ self-reported weight in kilograms divided by the square of their height in meters.

Self-Compassion: Participants completed Raes and colleagues’ short Self-Compassion Scale [79]. This measure is composed of 12 items, such as, “When I’m going through a very hard time, I give myself the caring and tenderness I need.” Items were rated on a 5-point Likert scale ranging from 1 (Almost never) to 5 (Almost always). In the present study, the Cronbach alpha was 0.90.

Body Esteem: Mendelson and colleagues’ Body Esteem Scale [30] was used to assess participants’ body esteem. This measure contains 23 items, such as, “I like what I see when I look in the mirror”. Items were rated on a 5-point Likert scale ranging from 1 (Never) to 5 (Always). In the present study, the Cronbach alpha was 0.95.

Diet Quality: Diet quality was assessed using a mean of food intakes from a web-based validated 24-h dietary recall, the R24W [80,81], to calculate the Canadian Healthy Eating Index (HEI-C) score. The score is composed of eleven components, which includes three moderation components, i.e., foods that should be limited in a healthy diet (saturated fats, sodium, and “other food”) and eight adequacy components, i.e., food items that should be consumed in greater amounts in order to achieve a healthy diet (total vegetables and fruit, whole fruit, dark green and orange vegetables, total grain products, whole grains, meat and alternatives, and unsaturated fats) [82]. The maximum possible score is 100, with moderation components contributing to 40 points and adequacy components to 60 points.

Intuitive Eating: Participants completed the 23-item intuitive eating scale [83], adapted and validated for the French-Canadian population [84]. A sample item is “I trust my body to tell me when to eat.” Items were rated on a 5-point Likert scale ranging from 1 (Completely disagree) to 5 (Completely agree). In the present study, the Cronbach alpha was 0.93.

Emotional Eating: The three-item emotional eating subscale from the Three Factor Eating Questionnaire (TFEQ-R18; [85,86]) was used to assess the participant’s propensity for emotional eating. A sample item is “When I feel lonely, I console myself by eating.” Items were rated on a 4-point Likert scale ranging from 1 (Definitely false) to 4 (Definitely true). In the present study, the Cronbach alpha was 0.90.

### 2.3. Statistical Analysis 

Descriptive analysis and Pearson’s correlations were conducted with IBM SPSS v.21 [87]. Descriptive analysis included calculating the mean and standard deviation for all the study variables (i.e., self-compassion, body esteem, diet quality, intuitive eating, and emotional eating) as well as participants’ age, BMI, number of children, and age of children. Pearson’s correlations, among all the aforementioned variables, were also computed. A *p*-value < 0.05 was considered statistically significant. Structural equation modeling analyses were conducted using AMOS 24 [88]. Model adequacy was estimated with the following fit indices: Normed Fit Index (NFI), Goodness of Fit Index (GFI), Adjusted Goodness of Fit Index (AGFI), Comparative Fit Index (CFI), Standardized Root Mean Square (SRMR), and Root Mean Square Error of Approximation (RMSEA). The cut-off values for those fit indices were NFI > 0.90, GFI > 0.90, AGFI > 0.90, CFI > 0.93, SRMR < 0.10, and RMSEA < 0.05 [89]. The estimated model was composed of three exogenous variables (i.e., age, BMI, and self-compassion) and four endogenous variables (i.e., body esteem, diet quality, intuitive eating, and emotional eating). The error terms of the three eating-behaviour outcome variables were allowed to covary.

Finally, the significance of the mediations between self-compassion and the three eating-behaviour outcome variables through body esteem were estimated with a bias-corrected bootstrapped 95% confidence interval (CI) estimate of indirect effects (see [90,91]) using AMOS 24 [88]. This technique allows for a statistical estimate of the indirect effects and their associated 95% confidence intervals by resampling the dataset multiple times in order to create several simulated samples. Past research suggests that the bias correction improves power and reduces Type 1 error rates [92]. For the present analysis, 1,500 simulated samples were created [93].

## 3. Results

### 3.1. Preliminary Analyses

Table 1 shows the means and standard deviations as well as the Pearson’s correlations between all study variables. The five main study variables (i.e., self-compassion, body esteem, diet quality, intuitive eating, and emotional eating) were found to be significantly inter-correlated. BMI was found to be significantly related to three of the five study variables and was thus included as a confounding variable in the path model. Participants’ age was marginally related to self-compassion and was thus controlled for in the analyses. The number of children was not significantly related to any of the study variables. Participants’ body esteem was found to be significantly related to the age of their eldest child. Yet, partial correlations revealed that body esteem was no longer related to the eldest child’s age once participants’ own age was controlled for. Therefore, the number of children and age of children were not included in the path analysis.

### 3.2. Main Analyses 

We calculated a structural equation model in which body esteem was modelled as a mediator between self-compassion and the three eating-behaviour outcome variables (diet quality, intuitive eating, and emotional eating). These paths were estimated while considering the associations of BMI with all the study variables. Age was included as a control variable. This first model was found to have an inadequate fit to the data: χ2 (df = 7, *n* = 100) = 8.845, *p* = 0.264, NFI = 0.972, CFI = 0.994; GFI = 0.976; AGFI = 0.904, RMSEA = 0.052 (0, 0.141), SRMR = 0.0408. Inspection of the modification indices suggested that the addition of a direct path from self-compassion to intuitive eating as well as a direct path from self-compassion to emotional eating would improve the fit of the model. This final model was found to have a satisfactory fit to the data: χ2 (df = 5, *n* = 100) = 1.501, *p* = 0.913, NFI = 0.995, CFI = 1.00; GFI = 0.996; AGFI = 0.976, RMSEA = 0 (0.00, 0.053), SRMR = 0.017, and was thus retained (see Figure 2).

First, a significant path was found between self-compassion and body esteem (β = 0.544, 95% CI (0.407; 0.669), *p* < 0.01). In turn, body esteem was found to be significantly related to diet quality (β = 0.335, 95% CI (0.070; 0.530), *p* < 0.01), intuitive eating (β = 0.583, 95% CI (0.398; 0.739), *p* < 0.01), and emotional eating (β = −0.361, 95% CI (−0.555; −0.129), *p* < 0.01). Self-compassion was also found to be significantly related to emotional eating (β = −0.248, 95% CI (−0.438; −0.044), *p* < 0.05) and marginally related to intuitive eating (β = 0.164, 95% CI (−0.023; 0.323), *p* = 0.080). All these effects were found while controlling for BMI, which was directly and significantly related to body esteem (β = −0.447, 95% CI (−0.544; −0.319), *p* < 0.01), intuitive eating (β = −0.165, 95% CI (−0.357; −0.008), *p* < 0.05), and emotional eating (β = 0.284, 95% CI (0.143; 0.452), *p* < 0.01), but not significantly related to diet quality (β = 0.038, 95% CI (−0.191; 0.223), *p* = 0.805). Results of the bootstrap analyses further revealed that body esteem significantly mediated the associations between self-compassion and diet quality, intuitive eating, and emotional eating (see Table 2).

## 4. Discussion

The present study sought to examine the role of body esteem in the associations between self-compassion, diet quality, and eating behaviours in a sample of mothers. The results showed, in line with our hypotheses, that higher levels of self-compassion were related to a higher diet quality, greater levels of intuitive eating, and lower levels of emotional eating. The results further showed that body esteem (at least partially) mediated these relations. Specifically, mothers with higher levels of self-compassion were found to have higher levels of body esteem, which, in turn, was related to a higher diet quality, greater levels of intuitive eating, and lower levels of emotional eating. The present study thus uncovered an important mechanism, namely body esteem, in explaining how mothers’ self-compassion relates to both eating behaviours and diet quality.

Body esteem, which refers to our self-evaluation regarding our appearance and weight, has previously been associated with self-care behaviours, healthy eating habits, and intuitive eating [26,27,28,29,32,33,34,35]. Conversely, recent studies have associated lower body esteem with unhealthy behaviours such as unhealthy diets, emotional eating, and lower levels of exercise [8,12,37,94]. 

Replicating previous work, we found that body esteem mediated the associations of self-compassion with eating behaviours [12,22,95]. More precisely, our results show that a higher level of self-compassion was related to higher body esteem which was, in turn, related to higher diet quality, greater intuitive eating, as well as lower emotional eating. The present results thus offer a replication of previous findings regarding emotional eating and intuitive eating, as well as add to them by extending the mediation to consider the association between self-compassion and diet quality. A better understanding of the associations between these variables in a sample of mothers is especially valuable given that children observe and imitate how and what their parents eat, as well as how they talk about their bodies (e.g., [13,78]).

The indirect association of self-compassion with diet quality through body esteem was found to be significant, which suggests a complete mediation through body esteem. However, while the indirect relationships between self-compassion and intuitive eating and emotional eating were significant, self-compassion was also found to have a significant direct negative association with emotional eating and a marginally significant direct positive association with intuitive eating. It therefore appears that treating oneself with kindness and being aware of one’s emotions and attitudes acts as a direct protective factor against emotional eating. Since emotional eating occurs when one uses food as a coping mechanism, for example, as a response to a negative emotional state [48,49,52,53], increased self-compassion may offer individuals self-care tools that help them rely less on food as a coping strategy. This is promising given that recent intervention studies have shown that increasing participants’ self-compassion can lead to improved eating behaviours and body image outcomes (e.g., [96,97,98,99,100]).

### 4.1. Implications

The present research has several implications for those hoping to target body esteem and eating behaviours through self-compassion. Numerous interventions as simple as exposure to quotes about self-compassion on social media [99], writing a short letter to oneself expressing kindness, compassion and understanding for one’s weight and appearance [101], or more in-depth interventions (e.g., [96,97,98,99,100]) appear to have positive impacts on eating behaviours and body image. Self-compassion can also be induced by sharing messages of self-kindness, mindfulness, and common humanity. For example, in an experimental study by Adams and Leary [94], in which participants were asked to eat unhealthy food, those in the self-compassion intervention condition were encouraged to not feel guilty or be hard on themselves after consuming the unhealthy food, and were reminded that everyone eats unhealthily sometimes. Thus, research suggests self-compassion is a malleable ability that can be enhanced via relatively simple and cost-effective interventions. This has important applied implications for counselors, nutritionists, and other allied health professionals who want to enhance their clients’ body esteem and can make use of this lever for change. The research of the present study speaks to the potential for improvements in self-compassion being associated with healthier eating behaviours. Designing interventions to help mothers be more gentle and less critical of themselves would be valuable, especially for mothers who have unrealistic views of what the “ideal mother” should eat and how she should look.

A second implication of this research is that a potential avenue for improving the body image and eating behaviours of children is the enhancement of self-compassion in mothers. Past research has established that mothers play a significant role in shaping their children’s eating behaviours and attitudes as well as in supporting their development of a positive body image [69,70,71,72]. For example, parents’ own weight and shape concerns have been shown to be transferable to their children [73,74,75,76] and females report lower levels of eating disturbances when they perceive their mother to communicate positive messages about weight and appearance [102]. Importantly, self-compassion appears to be modeled by mothers and can grow within the family environment (e.g., [5,103,104,105]). 

Thus, considering the important role that mothers play in transmitting attitudes and behaviours related to body image to the next generation, the present research points to a possible antecedent for promoting a positive relationship with food and body image in children and young adults (via mothers’ own self-compassion). Of course, future research is needed to confirm whether enhancing self-compassion in mothers also enhances the body image and eating behaviours of her children over time. 

### 4.2. Strengths and Limitations

One strength of the present study is the proposal of a mechanism through which self-compassion may be associated with diet quality and eating behaviours, namely, body esteem. However, this study is not without limitations. The inclusion of a non-clinical sample of women means that further research is needed to examine if these processes generalize to males and to clinical populations. Another limitation is that the present sample was highly homogenous. Participants were all White and reported a relatively high socio-economic status. Future research would benefit from a larger and more diverse sample. Additionally, the present study relied on self-reported data and used a cross-sectional design. Future studies would benefit from different designs, such as diary studies, where daily variations in self-compassion and body esteem could be studied in relation to diet quality and eating behaviours. Finally, all data were collected online, which results in a number of limitations such as self-selection bias, which may limit the generalizability of the present findings.

## 5. Conclusions

In sum, the present study highlighted the significant associations of self-compassion with women’s diet quality and eating behaviours and showed that these links occur, in part, due to improved body esteem. The present sample included the mothers of young children and of adolescents, which is important considering the paramount role they play in their children’s developing body image and eating behaviours. Future research is necessary, however, in order to build upon the present findings as well as to study the generalizability of the results in different samples (e.g., in males or in clinical samples) and to determine if self-compassion or body esteem can be enhanced through interventions in the hope of improving diet quality, increasing intuitive eating, and lessening emotional eating.

## Figures and Tables

**Figure 1 nutrients-13-03907-f001:**
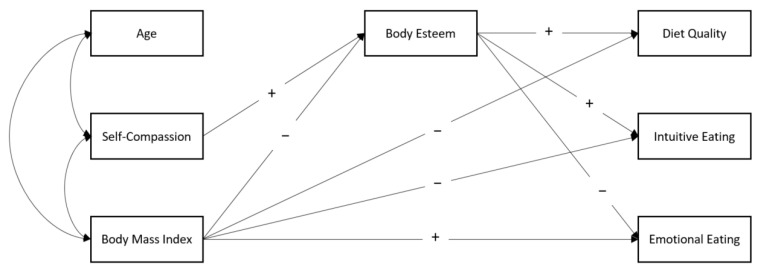
Hypothesized model. Note: The plus sign indicates that a positive link is expected; the minus sign indicates that a negative link is expected.

**Figure 2 nutrients-13-03907-f002:**
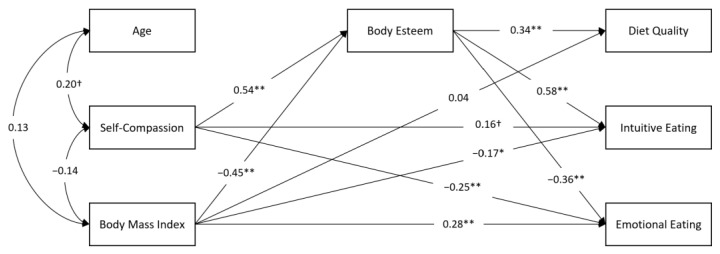
Results of the final path analysis model. Note: *n* = 100; ^†^ *p* < 0.10; * *p* < 0.05; ** *p* < 0.01.

**Table 1 nutrients-13-03907-t001:** Descriptive statistics and Pearson’s correlations for the study variables.

	Mean (SD)	Age of Mother	Age of Youngest Child	Age of Eldest Child	Number of Children	Mother’s BMI	Self-Compassion	Body Esteem	Diet Quality	Intuitive Eating
Age of Mother	33.70 (4.48)	-								
Age of Youngest Child	2.89 (1.85)	0.41 ***	-							
Age of Eldest Child	5.67 (2.91)	0.27 **	0.36 ***	-						
Number of Children	2.08 (0.87)	0.09	−0.19 †	0.55 ***	-					
Mother’s BMI	26.72 (7.72)	0.13	0.16	0.16	0.02	-				
Self-Compassion	3.12 (0.67)	0.20 †	0.09	−0.02	0.01	−0.14	-			
Body Esteem	3.18 (0.85)	0.05	−0.10	−0.20 *	−0.10	−0.52 ***	0.60 ***	-		
Diet Quality	58.90 (11.13)	−0.02	0.02	−0.06	0.03	−0.14	0.24 *	0.32 ***	-	
Intuitive Eating	3.65 (0.66)	0.08	−0.01	−0.15	−0.06	−0.49 ***	0.54 ***	0.77 ***	0.34 ***	-
Emotional Eating	2.34 (0.95)	−0.08	−0.07	0.09	0.15	0.51 ***	−0.51 ***	−0.66 ***	−0.23 *	−0.81 ***

Note: *n* = 100 mothers; † *p* < 0.10; * *p* < 0.05; ** *p* < 0.01; *** *p* < 0.001. BMI = body mass index. - indicates that a variable cannot correlate with itself.

**Table 2 nutrients-13-03907-t002:** Bias-corrected bootstrapped estimated of the mediations.

	Unstandardized Indirect Effect (Standard Error)	Bias-Corrected Bootstrapped 95% Confidence Interval Estimates	*p* Value
Self-compassion → Body esteem → Diet quality	3.03 (1.17)	(0.765; 5.336)	0.013
Self-compassion → Body esteem → Intuitive eating	−0.28 (0.09)	(−0.472; −0.106)	0.002
Self-compassion → Body esteem → Emotional eating	0.31 (0.07)	(0.187; 0.460)	0.001

Note: *n* = 100 women.

## Data Availability

The data are available on request to the corresponding author.

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
