# Peer review of "Feel Good, Eat Better: The Role of Self-Compassion and Body Esteem in Mothers’ Healthy Eating Behaviours"

_nutrients, 2021, doi:10.3390/nu13113907_

Round 1

Reviewer 1 Report

Thank you for the opportunity to review this manuscript. 

The manuscript is well written and would be an important addition to the literature. I have included some suggestions below for the authors to consider to further improve the manuscript 

Intro

  • A clearer link in the first section of the introduction between eating behaviours, body image and health in mothers
  • I recommend considering a diagram to show the relationships discussed in sections 1.1-1.4
  • I recommend rephrasing your 'present research'. This section is too simplified. You suggest that the only barrier to 'wanting to take care of themselves' is respect. However, it is so much more complex and multifaceted than that. Please consider reducing the strength of your language in this section or just using linked 176-179. 
  • The introduction needs to highlight the link between body image/respect and motherhood 

Methods: 

  • Please provide more information on the procedures ie. when was the sample recruited, how were they recruited, ethics approval, how was the data collected (online vs paper or face to face), where did you recruit the sample from? 
  • Where only 100 women recruited or was this the number included in the analysis?
  • Please provide more information about the statistical analysis. What variables were included in the descriptive and correlation analysis? How were they coded? What was the significance value for these analyses?
  • Please outline why you controlled for BMI, to me this seems outdated. We know that emotional eating, disordered eating, diet quality, self-compassion etc impact those across the BMI spectrum. I actually think it would be more helpful to control for diet quality. 

Results 

  • Presenting the Cronbach alpha in table one is not appropriate. This is a measure of internal consistency. These measures are not measuring the same domain and therefore a correlation coefficient would be more appropriate
  • Is there an association between number of children/child age and any of the outcomes?

Discussion: 

  • Please again consider the implications being a mother has on these outcomes 
  • Please discuss how/if self-compassion can be influenced 
  • Strengths and limitations need to be updated with the addition of further methodological reporting 

Reviewer 2 Report

Nutrients:

Feel Good, Eat Better: The Role of Self-Compassion and Body 2 Esteem in Mothers’ Healthy Eating Behaviours

The goal of the study was to examine associations between self-compassion, body esteem and a number of outcome variables related to eating behaviors (including emotional eating and intuitive eating). The results indicated that self-compassion is positively related to positive eating behaviors and negatively related to negative eating behaviors, and that these relationships may in part operate through body esteem levels. The study is tests hypotheses that are related to current topics of interest in therapy (self-compassion, dysregulated eating) and proposes a mechanisms of action. There are a few pieces of information that should be added to the Methods and Results sections; please find detailed feedback below.

Introduction:

  • Section 1.5: It would be helpful to list the specific hypotheses and instead of saying “we propose,” saying “we hypothesize.” If it was an exploratory study, that could be stated too, but based on the authors’ language, it appears that they made specific predictions about the directionality of the relationships they were studying.

Method:

  • 2.1: Where were the participants recruited from? It appears that the sample was not very diverse (perhaps it is representative of the area?..) and economically in middle class. Did the participants volunteer for the study or were the analyses part of another larger data collection. What were the inclusion and exclusion criteria?
  • 2.3: Please specify that those were Pearson r correlations.
  • If the PROCESS macro was used for bootstrapping, please specify. How many resamples were used (I assume the standard 5000 but it is helpful to explicitly state). Reporting the unstandardized coefficients is customary to indicate the relative strength of the mediation relationships.

Results

  • Table 1: The caption should also include mention of correlations (in addition to descriptives).
  • I think that the presentation of the bootstrapping analyses would benefit from figures reporting the unstandardized coefficients for each path.

Discussion

  • In the first paragraph, please state what you found instead of what you proposed (that will make it easier to follow) and you can add that these findings were consistent with the hypotheses.
  • “The present study uncovered an important mechanism explaining how self-compassion influences both eating behaviours and diet quality.” – I would replace this sentence with a sentence stating what that mechanism is to make the flow easier for the reader. It will also lead nicely into your next paragraph.
  • “Replicating previous work, we found that body esteem mediated the associations of self-compassion with diet quality and eating behaviours” – please state the directions of these relationships. For examples, “…such that higher self-compassion was associated with higher body esteem, which in turn was associated with higher diet quality,” etc.
  • What is a “self-compassion prime”? (ln 332) Can you explain more in depth how the participants were primed in that particular study?
